# Influence of COVID-19 confinement on students' performance in higher education

T. Gonzalez[1], M. A. de la Rubia[2], K. P. Hincz[3], M. Comas-Lopez[3], Laia Subirats[4,5], Santi Fort[4], G. M. Sacha[3]*

**1** Biochemistry Department, Faculty of Medicine, Universidad Autónoma de Madrid, Madrid, Spain, **2** Chemical Engineering Department, Faculty of Sciences, Universidad Autónoma de Madrid, Madrid, Spain, **3** Escuela Politécnica Superior, Universidad Autónoma de Madrid, Madrid, Spain, **4** Eurecat, Centre Tecnològic de Catalunya, Barcelona, Spain, **5** ADaS Lab, Universitat Oberta de Catalunya, Barcelona, Spain

* sacha.gomez@uam.es

**Data Availability Statement:** All relevant data are within the manuscript and its Supporting Information files.

## Abstract

This study analyzes the effects of COVID-19 confinement on the autonomous learning performance of students in higher education. Using a field experiment with 458 students from three different subjects at Universidad Autónoma de Madrid (Spain), we study the differences in assessments by dividing students into two groups. The first group (control) corresponds to academic years 2017/2018 and 2018/2019. The second group (experimental) corresponds to students from 2019/2020, which is the group of students that had their face-to-face activities interrupted because of the confinement. The results show that there is a significant positive effect of the COVID-19 confinement on students' performance. This effect is also significant in activities that did not change their format when performed after the confinement. We find that this effect is significant both in subjects that increased the number of assessment activities and subjects that did not change the student workload. Additionally, an analysis of students' learning strategies before confinement shows that students did not study on a continuous basis. Based on these results, we conclude that COVID-19 confinement changed students' learning strategies to a more continuous habit, improving their efficiency. For these reasons, better scores in students' assessment are expected due to COVID-19 confinement that can be explained by an improvement in their learning performance.

## 1. Introduction

The coronavirus COVID-19 outbreak disrupted life around the globe in 2020. As in any other sector, the COVID-19 pandemic affected education in many ways. Government actions have followed a common goal of reducing the spread of coronavirus by introducing measures limiting social contact. Many countries suspended face-to-face teaching and exams as well as placing restrictions on immigration affecting Erasmus students [1]. Where possible, traditional classes are being replaced with books and materials taken from school. Various e-learning platforms enable interaction between teachers and students, and, in some cases, national television shows or social media platforms are being used for education. Some education systems announced exceptional holidays to better prepare for this distance-learning scenario.

**Funding:** This research was funded by ADeAPTIVE (Advanced Design of e-Learning Applications Personalizing Teaching to Improve Virtual Education) project with the support of the Erasmus + programme of the European Union (grant number 2017-1-ES01-KA203-038266). This study was also funded by ACCIÓ, Spain (Pla d'Actuació de Centres Tecnològics 2019) under the project Augmented Workplace. This study was also funded by the Fondo Supera COVID-19 (Project: Development of tools for the assessment in higher education in the COVID-19 confinement).

**Competing interests:** The authors have declared that no competing interests exist.

In terms of the impact of the COVID-19 pandemic on different countries' education systems many differences exist. This lack of homogeneity is caused by such factors as the start and end dates of academic years and the timing of school holidays. While some countries suspended in-person classes from March/April until further notice, others were less restrictive, and universities were only advised to reduce face-to-face teaching and replace it with online solutions wherever practicable. In other cases, depending on the academic calendar, it was possible to postpone the start of the summer semester [2].

Fortunately, there is a range of modern tools available to face the challenge of distance learning imposed by the COVID-19 pandemic [3]. Using these tools, the modification of contents that were previously taught face-to-face is easily conceivable. There are however other important tasks in the learning process, such as assessment or autonomous learning, that can still be challenging without the direct supervision of teachers.

All these arguments end in a common topic: how to ensure the assessment's adequacy to correctly measure students' progress. Thus, how can teachers compare students' results if they differ from previous years? On one hand, if students achieve higher scores than in previous years, this could be linked with cheating in online exams or with changes in the format of the evaluation tools. On the other hand, lower grades could also be caused by the evaluation format change or be attributable to autonomous learning as a less effective teaching method.

The objective of this article is to reduce the uncertainty in the assessment process in higher education during the COVID-19 pandemic. To achieve this goal, we analyze students' learning strategies before and after confinement. Altogether, our data indicates that autonomous learning in this scenario has increased students' performance and higher scores should be expected. We also discuss the reasons underneath this effect. We present a study that involves more than 450 students enrolled in 3 subjects from different degrees from the Universidad Autónoma de Madrid (Spain) during three academic years, including data obtained in the 2019/2020 academic year, when the restrictions due to the COVID-19 pandemic have been in force.

## 1.1. Background

E-learning has experienced significant change due to the exponential growth of the internet and information technology [4]. New e-learning platforms are being developed for tutors to facilitate assessments and for learners to participate in lectures [4, 5]. Both assessment processes and self-evaluation have been proven to benefit from technological advancement. Even courses that solely offer online contents such as Massive Open Online Courses (MOOCs) [6, 7] have also become popular. The inclusion of e-Learning tools in higher education implies that a greater amount of information can be analyzed, improving teaching quality [8–10]. In recent years, many studies have been performed analyzing the advantages and challenges of massive data analysis in higher education [11]. For example, a study of Gasevic et al. [12] indicates that time management tactics had significant correlations with academic performance. Jovanovic et al also demonstrated that assisting students in their management of learning resources is critical for a correct management of their learning strategies in terms of regularity [13].

Within few days, the COVID-19 pandemic enhanced the role of remote working, e-learning, video streaming, etc. on a broad scale [14]. In [15], we can see that the most popular remote collaboration tools are private chat messages, followed by two-participant-calls, multi-person-meetings, and team chat messages. In addition, several recommendations to help teachers in the process of online instruction have appeared [16]. Furthermore, mobile learning has become an alternative suitable for some students with fewer technological resources. Regarding the feedback of e-classes given by students, some studies [17] point out that students

were satisfied with the teacher's way of delivering the lecture and that the main problem was poor internet connection.

Related to autonomous learning, many studies have been performed regarding the concept of self-regulated learning (SRL), in which students are active and responsible for their own learning process [18, 19] as well as being knowledgeable, self-aware and able to select their own approach to learning [20, 21]. Some studies indicated that SRL significantly affected students' academic achievement and learning performance [22–24]. Researchers indicated that students with strongly developed SRL skills were more likely to be successful both in classrooms [25] and online learning [26]. These studies and the development of adequate tools for evaluation and self-evaluation of learners have become especially necessary in the COVID-19 pandemic in order to guarantee good performance in e-learning environments [27].

Linear tests, which require all students to take the same assessment in terms of the number and order of items during a test session, are among the most common tools used in computer-based testing. Computer adaptive test (CAT), based on item response theory, was formally proposed by Lord in 1980 [28–30] to overcome the shortfalls of the linear test. CAT allows dynamic changes for each test item based on previous answers of the student [31]. More advanced CAT platforms use personalization to individual learner´s characteristics by adapting questions and providing tailored feedback [32]. Research contains numerous examples of assessment tools that can guide students [33–35] and many advances have also been developed in the theoretical background of CAT [36]. In this aspect, advantages offered by CAT go beyond simply providing a snapshot score [37], as is the case with linear testing. Some platforms couple the advantages of CAT-specific feedback with multistage adaptive testing [38]. The use of CAT is also increasingly being promoted in clinical practice to improve patient quality of life. Over the decades, different systems and approaches based on CAT have been used in the educational space to enhance the learning process [39, 40]. Considering the usage of CAT as a learning tool, establishing the knowledge of the learner is crucial for personalizing subsequent question difficulty. CAT does have some negative aspects such as continued test item exposure, which allows learners to memorize the test answers and share them with their peers [41, 42]. As a solution to limit test item exposure, a large question bank has been suggested. This solution is unfeasible in most cases, since most of the CAT models already require more items than comparable linear testing [43].

## 1.2. Purpose

The aim of this study is to identify the effect of COVID-19 confinement on students' performance. This main objective leads to the first hypothesis of this study which can be formulated as H1: COVID-19 confinement has a significant effect on students' performance. The confirmation of this hypothesis should be done discarding any potential side effects such as students cheating in their assessment process related to remote learning. Moreover, a further analysis should be done to investigate which factors of COVID-19 confinement are responsible for the change.

A second hypothesis is H2: COVID-19 confinement has a significant effect on the assessment process. The aim of the project was therefore to investigate the following questions:

1. Is there any effect (positive or negative) of the COVID-19 confinement on students' performance?

2. Is it possible to be sure that the COVID-19 confinement is the origin of the different performance (if any)?

3. What are the reasons for the differences (if any) in students' performance?

4. What are the expected effects of the differences in students' performance (if any) in the assessment process?

## 2. Materials and methods

### 2.1 Measurement instruments

We have used two online platforms. The first one is e-valUAM [44], an online platform that aims to increase the quality of tests by improving the objectivity, robustness, security and relevance of assessment content. e-valUAM implements all the CAT tests described in the following sections. The second online platform used in this study is the Moodle platform provided by the Biochemistry Department from Universidad Autónoma de Madrid, where all the tests that do not use adaptive questions are implemented. Adaptive tests have been used in the subjects "Applied Computing" and "Design of Water Treatment Facilities". Traditional tests have been used in the subject "Metabolism".

**2.1.1 CAT theoretical model.** Let us consider a test composed by $N_Q$ items. In the most general form, the normalized grade $S_j$ obtained by a student in the j-attempt will be a function of the weights of all the questions $\boldsymbol{\alpha}$ and the normalized scores $\boldsymbol{\psi}$ ($S_j = S_j(\boldsymbol{\alpha}, \boldsymbol{\varphi})$), and can be defined as:

$$S_j(\overrightarrow{\alpha}, \overrightarrow{\varphi}) = \left(\sum_{i=1}^{N_Q} \alpha_i \varphi_i\right) \tag{1}$$

where the $\varphi_i$ is defined as

$$\varphi_i = \delta_{A_i R_i} \tag{2}$$

where $\delta$ is the Kronecker delta, $A_i$ the correct answer and $R_i$ the student's answer to the i-question. By using this definition, we limit $\varphi_i$ to only two possible values: 1 and 0; $\varphi_i = 1$ when the student's answer is correct and $\varphi_i = 0$ when the student gives a wrong value. This definition is valid for both open answer and multiple-choice tests. In the case of multiple-choice test with $N_R$ possible answers, $\varphi_i$ can be reduced to consider the random effect. In this case:

$$\varphi_i = \delta_{A_i R_i} - \frac{1 - \delta_{A_i R_i}}{(N_R - 1)} \tag{3}$$

Independently of using Eqs 2 or 3, to be sure that $S_j(\boldsymbol{\alpha}, \boldsymbol{\varphi})$ is normalized (i.e. $0 <= S_j(\boldsymbol{\alpha}, \boldsymbol{\varphi}) <= 1$), we must impose the following additional condition on $\boldsymbol{\alpha}$:

$$\sum_{i=1}^{N_Q} \alpha_i = 1 \tag{4}$$

In the context of needing a final grade (FG) between 0 and a certain value M, which typically takes values such as 10 or 100, we just need to rescale the $S_j(\boldsymbol{\alpha}, \boldsymbol{\varphi})$ value obtained in our model by a factor K, i.e. $FG_j = K\, S_j(\boldsymbol{\alpha}, \boldsymbol{\varphi})$.

We will now include the option of having questions with an additional parameter L, which will be related to the level of relevance of the question. L is a number that we will assign to all the questions included in the repository of the test (i.e. the pool of questions from where the questions of a j-test will be selected). The concept of relevance can take different significances depending on the context and the opinion of the teachers. In our model, the questions with lower L values will be shown initially to the students, when the students answer correctly a certain number of questions with the lower L value, the system starts proposing questions from the next L value. By defining $N_L$ as the number of possible L values, the L value that must be obtained in the k-question of the j-test can be defined as:

$$L_k = trunc\left[\left(\frac{N_L \sum_{i=1}^{k-1} \varphi_i}{N_Q}\right) + 1\right] \tag{5}$$

where *trunc* means the truncation of the value between brackets. It is worth noting that $L_k$ is proportional to the sum of the student's answers to all the previous questions in the test. This fact means that, in our model, the $L_k$ depends on the full history of answers given by the student. $L_k$ is inversely proportional to $N_Q$, which means that it takes a higher number of correct answers to increase $L_k$. Once $L_k$ is defined, a randomly selected question is shown to the student. Another important fact that implies the use of Eq 5 in the adaptive test is that we will never have $L_k < L_{k-1}$. In other words, once a student starts answering questions from a certain L value, they will never go back to the previous one. In Fig 1 we show a simple example of the multiple possible paths that a student can follow when facing a test that uses our model. In the example shown in Fig 1 we have used $N_Q = 6$ and $N_L = 3$. In this very straightforward example, the L value changes every time a student gives 2 correct answers (following diagonal arrows on the diagram). Wrong answers imply a vertical drop on the diagram. In this case, the L value will not change ($L_k = L_{k-1}$). The final student's grade G is shown on the lower part of the diagram. The lowest grade (G = 0) corresponds to a test where the student has failed all the $N_Q$ questions. In this case, L = 1 for all the possible j values. The highest grade (G = 6) implies that the student has faced questions from all the possible L values (L = {1,2,3} in Fig 1).

**2.1.2 Multiple Answer Test (MA-T).** MA-Ts are used for evaluation and self-evaluation of theoretical contents. In this case, we use a standard format where a single correct answer must be chosen from a pool of possible answers shown by the system. Since students can sometimes answer correctly by a random selection, Eq (3) must be used when evaluating the score of any item. The format of the MA-T in our CAT method requires the following elements:

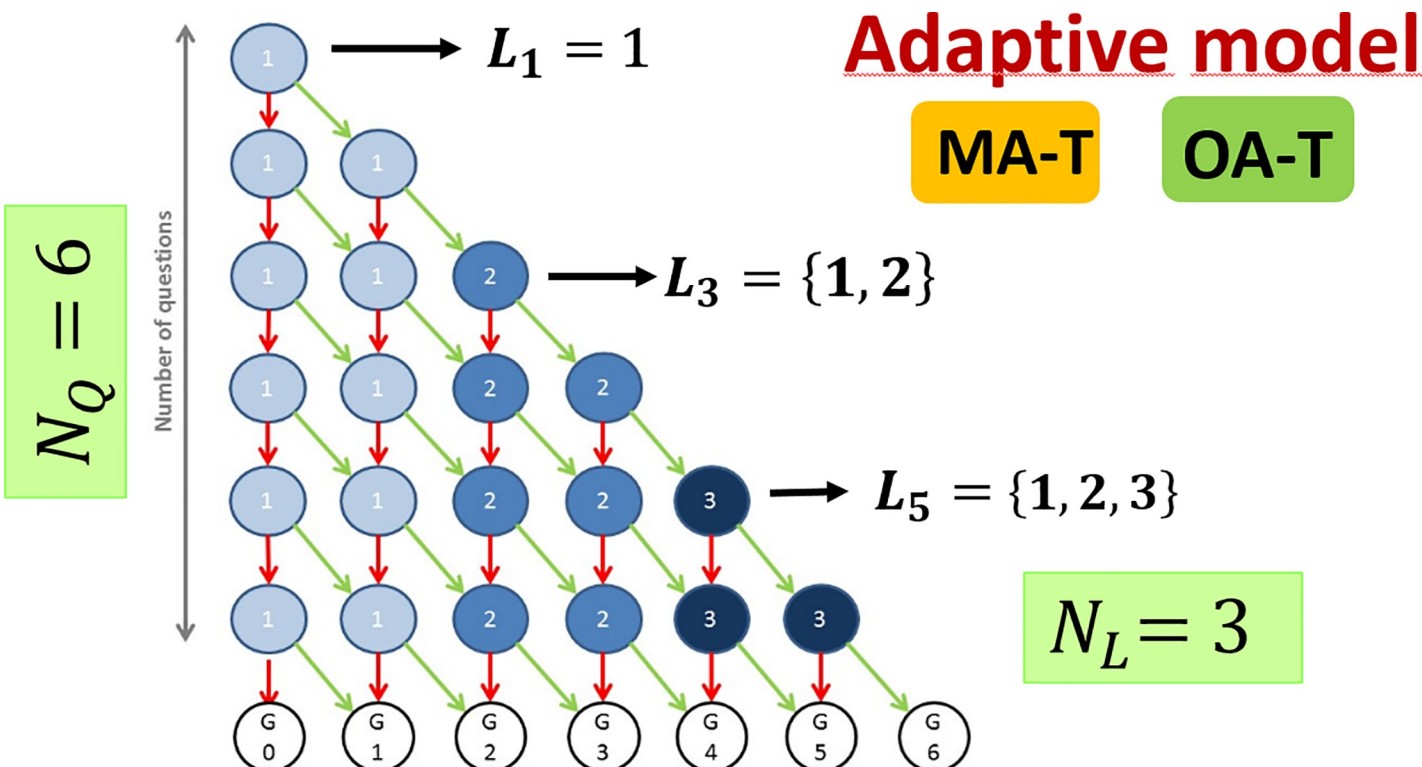

**Fig 1. Representation of the CAT model.** Green arrows indicate the path students take when they answer a question correctly. Students follow red arrows when an incorrect answer is given. Grades are represented at the bottom of the figure, improving with each step to the right.

- Statement

- Correct answer

- Rest of possible answers (wrong ones)

- Level of the item

In addition, the e-valUAM platform, where the method proposed in this article has been implemented, allows optional use of images or sounds for both the statement and the answers. e-valUAM shows the statement and the possible answers. All other information relating to the CAT method such as the level of the item is hidden from the student. The interface also shows optional feedback information about the performance of the learner in the previous item. Finally, it shows the time remaining to finish the test.

**2.1.3 Open Answer Test (OA-T).**   To train the contents of numerical problems, an OA-T was developed in which the statements include at least one parameter that will change its value with each execution of the application. This kind of question requires the following elements to be created:

- Statement with explicit indication of the modifiable parameter(s).

- Minimum and maximum values of each modifiable parameter.

- Programming code (Matlab in e-valUAM) that calculates the solution to the problem.

- Level of the item.

As with MA-T, there is also a possibility of including multimedia files in the item. However, in this type of question, the multimedia option is only available in the question as the answer is numerical and included by the user in all cases. In this case the interface has a space for entering the numerical answer. In this figure, we also show the feedback of the application when an incorrect previous answer has been introduced.

**2.1.4 Traditional tests.**   Traditional tests have been used in the subject "Metabolism" from the Human Nutrition and Dietetics Degree. The course contents are divided into 6 parts corresponding to different aspects of human metabolism. Each part is taught with face-to-face lectures followed by different on-line activities that the students perform in the Moodle platform and that are later discussed between them and with the professor in face-to-face workshops. All 6 parts of the course have a similar format, being a lecture followed by a selection of on-line activities in this order: "Exercises Workshop", "Discussion Workshop", "Self-assessment" and "Test". Except for the Discussion Workshops outlined below, all activities are on-line questionnaires (closed, multiple-choice questions) that the students must perform in Moodle, achieving a score that is automatically calculated and showed to them as part of their continuous assessment. Exercises Workshops and Self-assessments are completed by the students without tutor supervision, whereas the Tests are done on-line but under controlled conditions, just in the same way as a regular examination. Each student should perform a total of 15 on-line activities: 5 Exercises Workshops, 6 Self-assessment activities and 4 Tests. Discussion Workshops are practical cases that must be debated by the students and sent to the tutor on Moodle. As these activities do not obtain an automatic score but require a correction by the professor, they have not been considered in this study.

## 2.2 Design of the experiment

**2.2.1 Control group.**   The control group of our study is formed by the students of "Applied Computing" and "Metabolism" from academic years 2017/2018 and 2018/2019 and

by students of "Design of Water Treatment Facilities" from academic year 2017/2018. In the case of "Design of Water Treatment Facilities", a longitudinal study has been performed in academic year 2017/2018 to analyse the effect of rewards in the students' learning strategies, especially those related to time management.

**2.2.2 Experimental group.** The experimental group of our study are students of "Applied Computing" and "Metabolism" from academic year 2019/2020. In the longitudinal study of "Design of Water Treatment Facilities", experimental group corresponds to the third stage of the study.

**2.2.3 Study of autonomous learning strategies in the control group.** To answer our research questions, we first set up an experiment that obtains accurate measurements of the autonomous learning activities both in the control and the experimental groups. High accuracy in the measurements of the autonomous learning activities is achieved by using the previously described adaptive tools both in learning and assessment. In our experiment, the evaluation format and the e-learning tools are known to the students from the start, which motivates them to use the tools and perform the tests more consistently. In Fig 2 we show the procedure used in our experiments to measure autonomous learning. There are two kinds of contents in the subjects included in the experiment: theory and numerical problems. In the case of theory, students can use the e-valUAM platform through MA-T. The final evaluation is performed by OA-T. In the case of numerical problems, both learning and evaluation use OA-T. In this case, student motivation for using the platform is even higher because their final scores will be obtained by the same tool they use for their autonomous learning. This system is applied in all the experiments related to analysing the autonomous learning activity.

In the case of "Design of Water Treatment Facilities", an additional longitudinal study has been performed and is shown in Fig 2. This study has been performed in three stages. In the first stage, the self-evaluation material was presented to the students right before the evaluation test. In the second stage, the self-evaluation material was available 3 weeks before. To manage the theory contents, a MA-T with three possible answers to each question and only one correct has been created. In the first stage, with the test made available only one day before the exam, students had at their disposal a MA-T composed of 16 questions and with a time limit of 15 minutes. In the second stage, the time available was increased to 30 minutes and the question number to 18. These changes are related to the nature of the subject contents and do not have a significant impact on the study. The OA-T tests had 6 and 9 questions in the first and second stage, respectively. In order to maintain consistency in the subsequent study, an additional effort has been made to keep the difficulty of the questions as similar as possible.

Stage 3 had no new material added, instead, it used all the material created in stages 1 and 2. Stage 3 corresponded to the time window between the end of classes and the last exam. In this stage, all the material was available from the beginning. The protocol used in this stage only varied from stage 2 in the inclusion of a reward for students who used the application on 3 or more different days. The reward was related to a bonus in the final grades.

**2.2.4 Effect of COVID-19 confinement.** Once the correct measure of the autonomous learning is ensured, as explained in the previous section, we must develop an experiment to describe the effect of COVID-19 confinement. This experiment compares results of control and experimental groups in the whole assessment process. There are two stages that must be considered to determine the effect of the COVID-19 confinement. The first one corresponds to the period without confinement in the year 2019/2020 (before March 11), when all the measurable activities are performed in similar conditions for experimental and control groups. The second stage corresponds to the period of COVID-19 confinement (after March 11), where some measurable activities were performed in a different format and statistical differences can be found by comparing experimental and control groups.

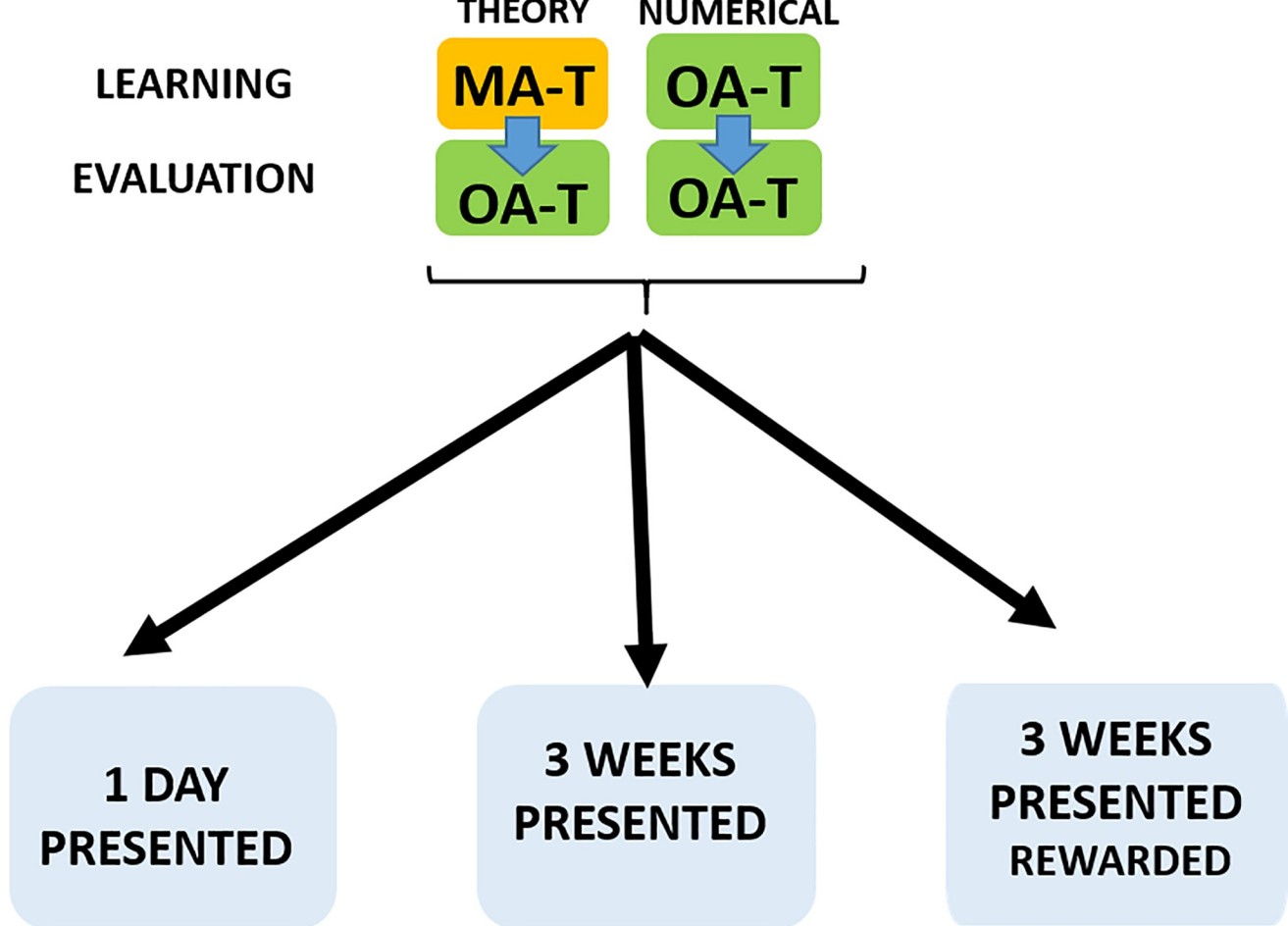

**Fig 2. Scheme of the strategy followed to accurately measure the autonomous work of students and the influence of rewards.**

In the case of "Applied Computing", students use the e-valUAM application on a continuous basis as it is available from the beginning of the course. They use the same OA-T for training and for their final exam, ensuring often and consistent use of the platform throughout the course. For this reason, in "Applied Computing", data analysis is performed over a continuous self-evaluation process over a single test along the whole academic year. The test performed by the students did not change its format or available questions in the whole three years under study.

In the case of "Metabolism", the confinement started just before the beginning of Part IV of the subject. Therefore, in this second stage, the scheduled in-person lectures from Part IV were recorded on video by the tutor and provided to the students. Afterwards, the students had an opportunity to perform the corresponding on-line continuous assessment activities, just in the same way as they would have done under normal conditions (as these activities were always

not face-to-face). As this subject has already had an important number of on-line activities programmed prior to the suspension of face-to-face teaching, it is an extremely valuable tool for analysing the effect of the confinement on students' performance.

In Fig 3, we show the full experiment described in this section. First, the two studies performed in years prior to COVID-19 confinement. In these first studies, we focus on the

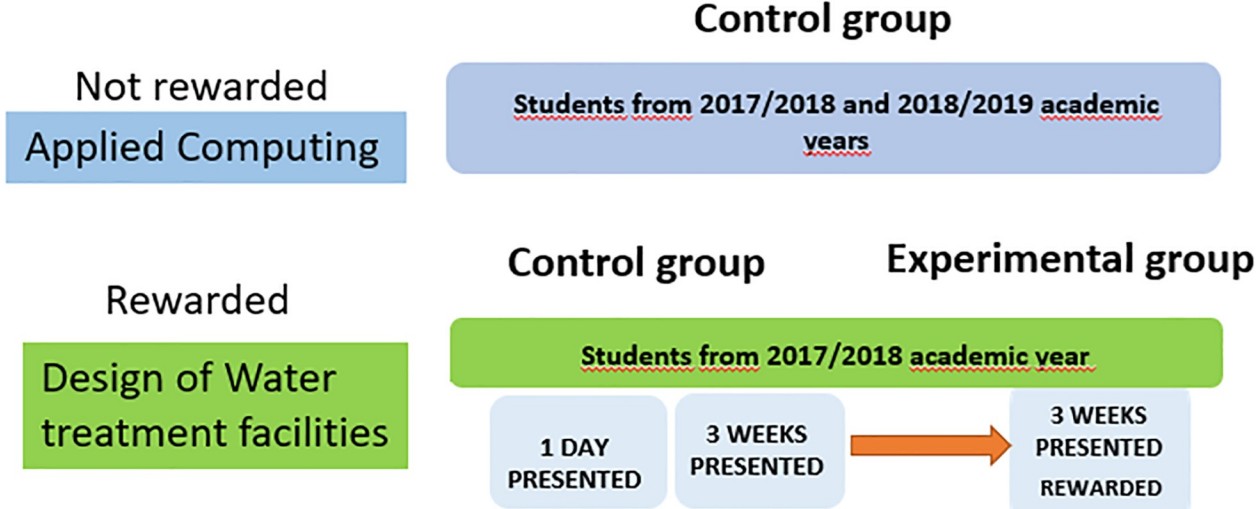

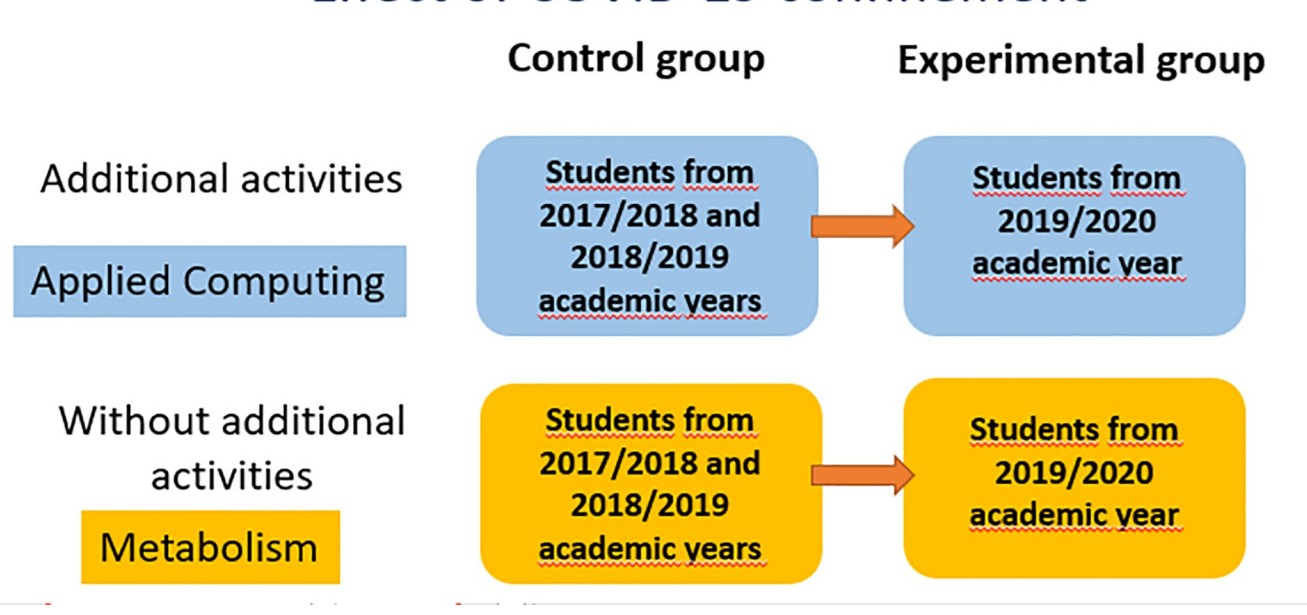

**Fig 3. Description of the experiment.**

learning strategies of the students and their response to reward scenarios. Information from these first studies will be useful for later discussion of the reasons for the changes caused by the COVID-19 confinement. The second part of our experiment is related to the effect of COVID-19 confinement, where we compare results of similar assessment activities between control and experimental groups. In Fig 3 we include information about all control and experimental groups in the different interventions.

## 2.3 Participants

This work involves data that has been collected from 458 students of the Universidad Autónoma de Madrid. This data has been collected in the following manner:

1. It does not involve minors.

2. It has been collected anonymously. Students have been identified by a numerical code, avoiding gathering of any personal information.

3. Students have been informed by the lecturers that some information about their activity could be anonymously collected for statistical purposes. Authors of this study did not receive any objections.

4. The tasks related to this study were completely voluntary and they did not in any form alter students' activities, classes, or the assessment process.

Considering these circumstances, we have received confirmation from the Research Ethics Committee of Universidad Autónoma de Madrid that we do not need to apply for ethics approval from our university since no personal data, minors or potentially hazardous activities were involved in the study.

Teachers involved in the study (coauthors of the present manuscript) who were responsible for the subjects taught also gave consent to carry out the study. We obtained verbal consent from the participants in the study.

Experiments have been performed in the subjects "Applied Computing", "Metabolism" and "Design of Water Treatment Facilities".

**2.3.1 Applied computing.**   This study with real students took place during the 2017/2018, 2018/2019 and 2019/2020 academic years in the first-year subject "Applied Computing". 97, 73 and 91 students were involved each academic year, respectively. This subject was taught through theory lessons and practical classes in the computer laboratory. This course corresponds to 6 ECTS and belongs to the Chemical Engineering Degree in the Faculty of Sciences from Universidad Autónoma de Madrid, Spain. Due to COVID-19 pandemic, the face-to-face teaching was cancelled on March 11, having a strong impact on the 2019/2020 course since it started in the first week of February and lasted until mid-May.

**2.3.2 Metabolism.**   The data for this study has been obtained from real students enrolled on the "Metabolism" course in the Human Nutrition and Dietetics Degree from Universidad Autónoma de Madrid, Spain. This is a 6 ECTS compulsory subject taught during the second semester of the first year of the degree. The study comprises data from 2017/2018 and 2018/2019 academic years, with 64 and 63 enrolled students, respectively, and from the present 2019/2020 academic year (47 students), which has been strongly affected by the restrictions caused by the COVID-19 pandemic. This year, the course started on January 28 and finished on April 30. Face-to-face teaching cancellation and beginning of confinement occurred approximately halfway through the semester.

**2.3.3 Design of water treatment facilities.**   The experiment with real students took place during the 2017/2018 academic year. 23 students were involved in the optional fourth-year

subject "Design of Water Treatment Facilities". This subject was taught through theoretical and practical classes in the classroom and laboratory, respectively. Our study focused on the theoretical part of the subject. This course corresponds to 6 ECTS and belongs to the Chemical Engineering Degree in the Faculty of Sciences of the Universidad Autónoma de Madrid, Spain.

## 2.4 Statistical analysis

**2.4.1 Applied computing.**   In this subject we have expressed student performance as a 0–10 score and analyzed it across all tests performed in self-evaluation. Those OA-T format tests were available from the beginning of the academic year and students used them on a continuous basis. Data from this subject imply results from an adaptive test. This imply that there are some points in the test where students get stacked and some scores are repeated more often than others. We tested if data followed a Gaussian distribution using a D'Agostino and Pearson omnibus normality test [45]. As the data did not pass the normality test, we used non-parametric statistical tests to compare the data from all 3 academic years. Means were compared by a Kruskal-Wallis test [46] and, when differences were found, by a Mann-Whitney post hoc test [47] to determine which pairs of means were different. All statistical analysis was performed using GraphPad Prism 6 (GraphPad Software, La Jolla, California, USA). Data is presented as mean±SD and statistical significance was set at $p < 0.05$.

**2.4.2 Metabolism.**   Here, we have analysed 2 variables, the score (0–10) obtained by the students in the different activities and the proportion of students that passed the activity (score≥5), from 2019/2020 and the 2 previous academic years. Firstly, we checked if the results from the previous 2 years (2017/2018 and 2018/2019) were similar, thus allowing them to be compared with results from the present year. To perform the analysis, we tested the score data for normality using a D'Agostino and Pearson omnibus normality test [41], later comparing them with an unpaired 2-tailed t-test, if they were normal, or with a non-parametric Mann-Whitney test, if they were not normal. Afterwards, we proceeded similarly to compare the results from the present year with those from the previous 2 years. When data from each activity followed a normal distribution, we compared the 3 means by 1-way ANOVA followed by an unpaired 2-tailed t-test. When the data did not follow a normal distribution, we compared the 3 means with a Kruskal-Wallis test followed by a Mann-Whitney post hoc test. In order to compare the proportion of students that passed the activities (score≥5) in different years, we performed a z-test (after confirming that our data met the central limit theorem). All statistical analysis was performed using GraphPad Prism 6. Data is presented as mean±SD and statistical significance was set at $p < 0.05$.

**2.4.3 Design of water treatment facilities.**   In order to substantiate related samples of results in our experiment, we performed a Wilcoxon Signed Rank test which compares two related samples that cannot be assumed to be normally distributed. In the experiments related to this subject, the same students were involved in both samples. Data is presented as mean ±SD and statistical significance was set at $p < 0.05$.

## 3. Results

### 3.1 Autonomous learning strategies in the control group

**3.1.1 Continuous working of students in a non-rewarded scenario.**   In this section, we analyze students' self-learning strategies in the subject "Applied Computing" in the years that were not influenced by COVID-19 (2017/2018 and 2018/2019). However, since the data is very similar in both courses, we will focus con 2017/2018 for clarity. All statistical results can be also applied to 2018/2019. In both years, theoretical lessons were taught in the classroom

and practical lessons in computer laboratories with the teacher physically present. Practical lessons always used the e-valUAM platform and the adaptive test described previously.

In Fig 4A, we show the scores obtained by all 2017/2018 students in all the attempts made with adaptive tests. Attempts are ordered chronologically. A score of -1 in the figure implies that the student did not finish the test and did not obtain a numerical score. Those tests have been included in the figure as a part of student's autonomous work and must be considered as learning time; however, we will exclude them from score data analysis. There are some patterns in the figure that must be considered. First, certain scores repeat more than others (2.5, 5 and 7.5), which corresponds to a jump in question level (L) (level 1: 0–2.5, level 2: 2.5–5, level 3: 5–7.5 and, level 4: 7.5–10), meaning that students were not yet able to answer more difficult questions. Another pattern that can be easily seen in the figure is a region where a huge number of tests had a score of -1 (between 600 and 700 approximately). On those days teachers asked the students to find the most difficult questions in the test. For that reason, students interrupted the tests when they reached those questions.

However, the most interesting pattern, which can be observed in Fig 4A, is the distribution of the tests in time, with a much bigger number of tests done in May, being the last month of the academic year for this subject. Students used the platform 688 times in total in May and only 486 times in the period between February and April. It is worth noting that the academic year for this subject ended with the final exam on May 21. In Fig 4B we show the distribution of attempts between February and May. We found 62 in February, which is a low but reasonable number since students did not yet have enough knowledge to face tests properly. They generally started using the platform in the middle of the month. In March, we noted an increase in the number of attempts followed by a slight decrease in April. This difference can be justified by the Easter holidays (one week) in April. The number of attempts strongly increased in May. Even when counting only the first two weeks of the month, we have more attempts than in previous months (292 from May 1–15). In the period of May 16–21, we have recorded 396 attempts (469 if we count the attempts in the final exam). In the inset of Fig 4B we show the attempt distribution in the last period (May 16–21). As we can see, students used the platform 127 times the day before the exam and 173 times the day of the final exam (95 for studying and 78 in the exam itself). More than 33% of tests were performed in the last 6 days before the final exam (and more that 50% of those tests were performed the day before the exam).

**3.1.2 Continuous working of students in a rewarded scenario.** To study the effects of a rewarded scenario, we have developed an experiment in the academic year 2017/2018 for students from the subject "Design of Water Treatment Facilities". We have used 3 stages (fully described in the Materials and methods section): stage 1, where the self-evaluation tests have been made available for 1 day; stage 2, where the self-evaluation tests have been made available for 3 weeks; and stage 3, where self-evaluation tests were available for a longer period with a reward related to their use. Fig 5 shows the attempts distribution in all the stages. Focusing on the MA-T, 13/20 students made more attempts in the first stage than in the second, which is striking considering that in the second stage application was available for longer. Three students have not made any attempts in either of the 2 stages.

As for the group of students who have used the application for more than one day, we noted only 3 individuals working for 3 days or more. Considering that the application was available for 3 weeks in this second stage, these results show the very low willingness of the students to work continuously. Focusing on the OA-T, this effect is even more significant, as only two cases in which students used the application for more than one day were recorded. Increasing student motivation and analyzing the results of a more persistent use of the tool was the objective of stage 3.

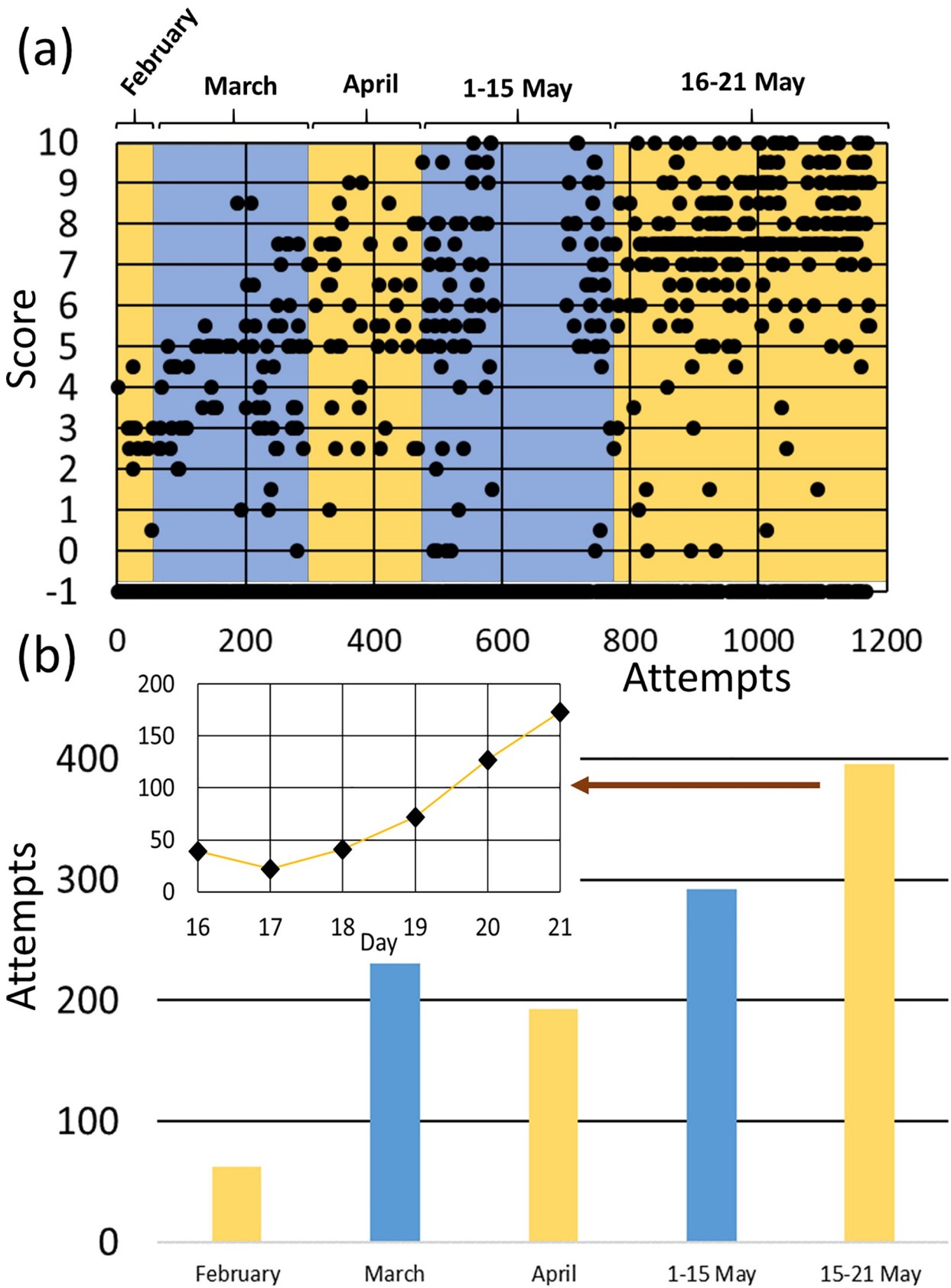

Fig 4. **Distribution of self-evaluation activities in "Applied Computing".** (a) Individual scores of all the students in the academic year 2017/2018 (ordered chronologically). Each dot corresponds to a single attempt of a student. (b) Number of attempts in the academic year 2017/2018 distributed in time periods. Inset shows the distribution in the days between May 16 and May 21.

In Fig 5 we can see the distribution of attempts made by the students in the whole study (3 stages). There is a fourth stage in the figure that corresponds to an additional exam taken by the students that did not pass the previous ones. Since only a few of them were involved in this fourth stage, it was excluded from the analysis. As we can see in the figure, attempts from stage 1 were limited to a couple of days, which corresponds to the short period when the test was available. Some attempts can be found in a wider window in stage 2. However, a much higher number of attempts extended in time can be found in stage 3, which corresponds to the rewarded stage.

In order to measure the use of the tool over time, we summarized the use of the system by students in stage 2 and 3 by comparing the number of days the students attempted the stage 2 MA-T and OA-T tests during the 3 weeks leading up to the exams given after each stage. It is interesting to note that, during stage 3, six more students utilized the system than in stage 2; however, the total attempts reduced between the 2 stages from 123 to 102. This translates to a

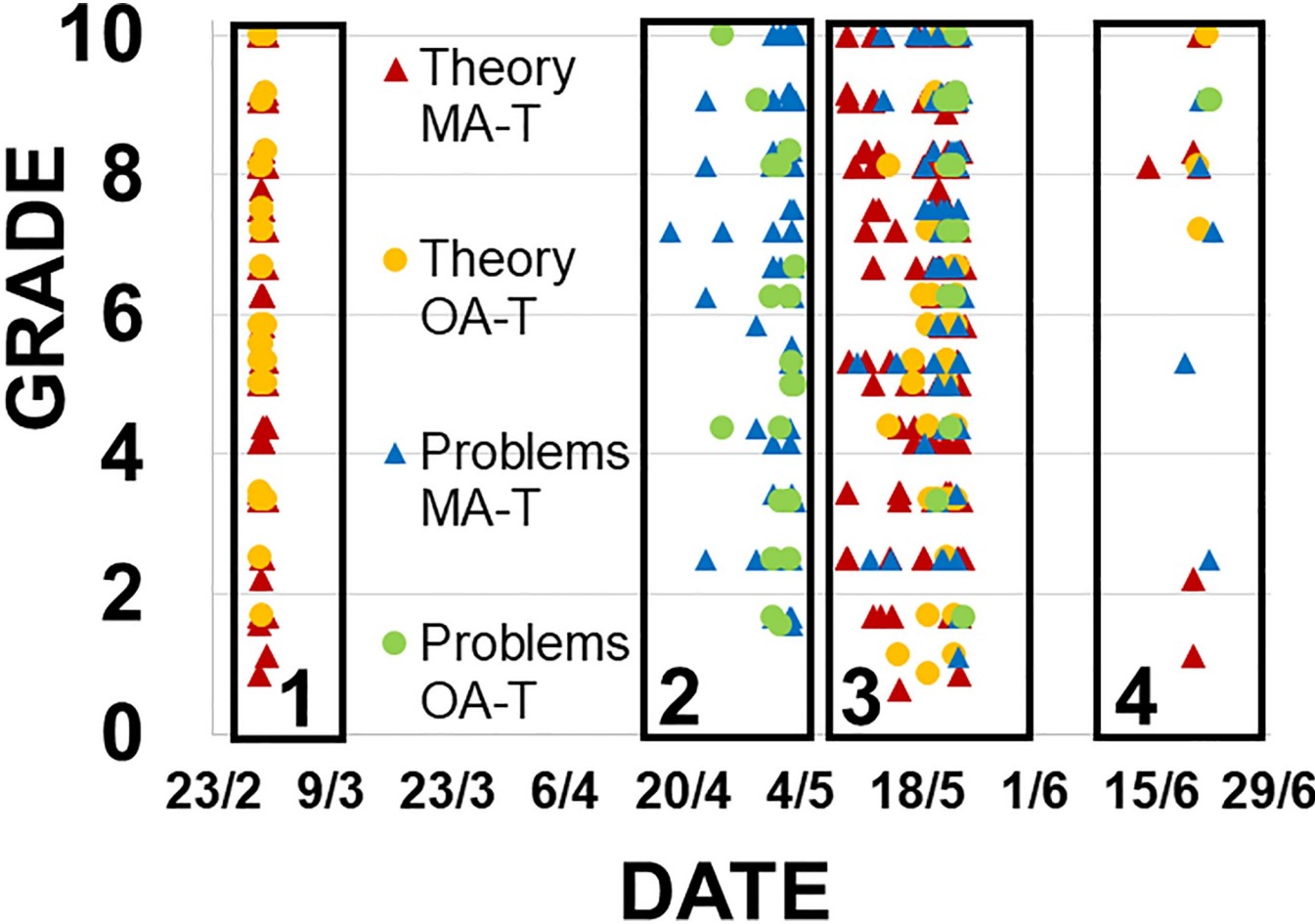

Fig 5. **Distribution of attempts in the self-evaluation/learning phase in "Design of Water Treatment Facilities".** The horizontal axis corresponds to the dates and the vertical axis to the grades obtained in every individual attempt. The attempts made by all students in the 3 stages are presented.

23% increase in the average use of the application over time from 1.9 days in stage 2 to 2.4 days in stage 3 combined with a reduction of attempts per student from 9.5 in stage 2, to 5.4 in stage 3. In the earlier stage, fewer students used the application intensively by making multiple attempts a day. Students who logged in at the later stage were more likely to use the application once or twice a day over several days because of the reward related to this behavior. Out of a total sample of 19 students who used the tool in stage 3, 5 were eliminated by the test as they accessed the tests the same number of days in both stages. Then, for the use of the application by the remaining 14 students the W-value was of 15.5, with the critical value of W = 25 at p.0.05. Therefore, the result is significant.

## 3.2 Effect of COVID-19 confinement

We have studied two cases related to two different and common scenarios in COVID-19 distance learning. The first one is related to subjects where teachers increased the number of tasks that students should perform autonomously. The second one is related to subjects where all the autonomous tasks were the same as in previous courses. In this case, only face-to-face sessions were replaced by distance learning activities such as online sessions or recorded classes.

**3.2.1 Additional autonomous activities during confinement.** In Fig 6 we show the scores obtained by all the students in the 3 courses analysed in all the e-valUAM tests performed in the subject "Applied Computing". In the figure we indicate the period of the COVID-19 confinement during the 2019/2020 course (from March 11 to March 30). For clarity, we also indicate the equivalent period in the two previous years. As we can see in the figure, the number of tests performed before confinement was similar (225 and 246) in the two previous years (control group), but there is a remarkable increment in the number of tests (328) performed in the experimental group. These differences can be easily explained normalizing these numbers to the number of students each year (97, 73 and 91 respectively). By doing that, we obtain 2.31, 3.37 and 3.6 test/student, respectively. Thus, differences between courses 2018/2019 and 2019/2020 are reduced, although there is still some difference with respect students from course 2017/2018, that can be explained by the fact that some of these students were retaking the subject and, therefore, did not use the e-valUAM application often until the contents were more difficult. If we count the number of tests finished until the end of the academic year in courses 2017/2018 and 2018/2019, we find 1345 and 1175 tests respectively. Normalizing with the number of students, we find 13.87 and 16.1 test/student respectively. This is a difference of 2.23 test/student in a period of 3.5 months, which means a difference of less than 1 test/student each month.

The conditions after the COVID-19 confinement radically changed in the course 2019/2020 because students were required to perform a higher number of tests each week. As we can see in Fig 6, there are still small differences between courses 2017/2018 and 2018/2019. However, the number of tests taken in 2019/2020 is almost 5 times higher. In the inset of Fig 6 we show a histogram with the data for course 2019/2020. Adaptive tests used in this experiment induce the effect of higher number of attempts at scores 2.5, 5 and 7.5, which are the points where the level of the questions increases. For that reason, the curve is not normal. This fact is statistically tested in the following analysis.

When comparing the mean scores from the period before confinement (Fig 7), we found that they were statistically different between the 3 academic years. Differences between the mean score from the 2019/2020 (experimental group) and both the other 2 years (control group) were: 4.5±1.6 in experimental group, vs. 3.9±1.5 in 2017/2018, p = 0.0003, and vs. 3.9 ±1.5 in 2018/2019, p = <0.002), with no differences between 2017/2018 and 2018/2019 (p = 0.997). Similarly, in the period between the beginning of confinement and March 30,

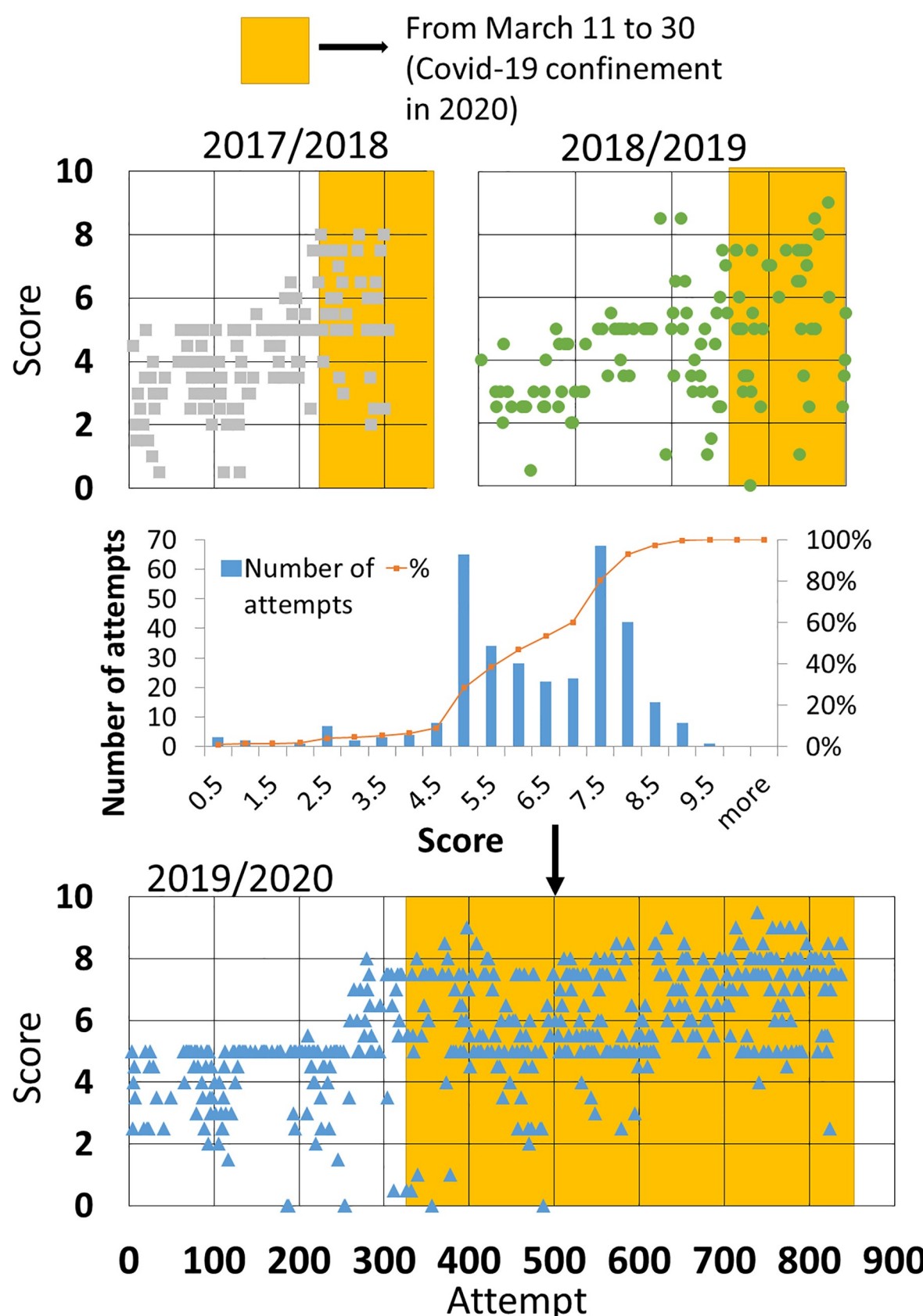

**Fig 6. Attempts vs. scores ordered chronologically for the three academic years under study in "Applied Computing".**

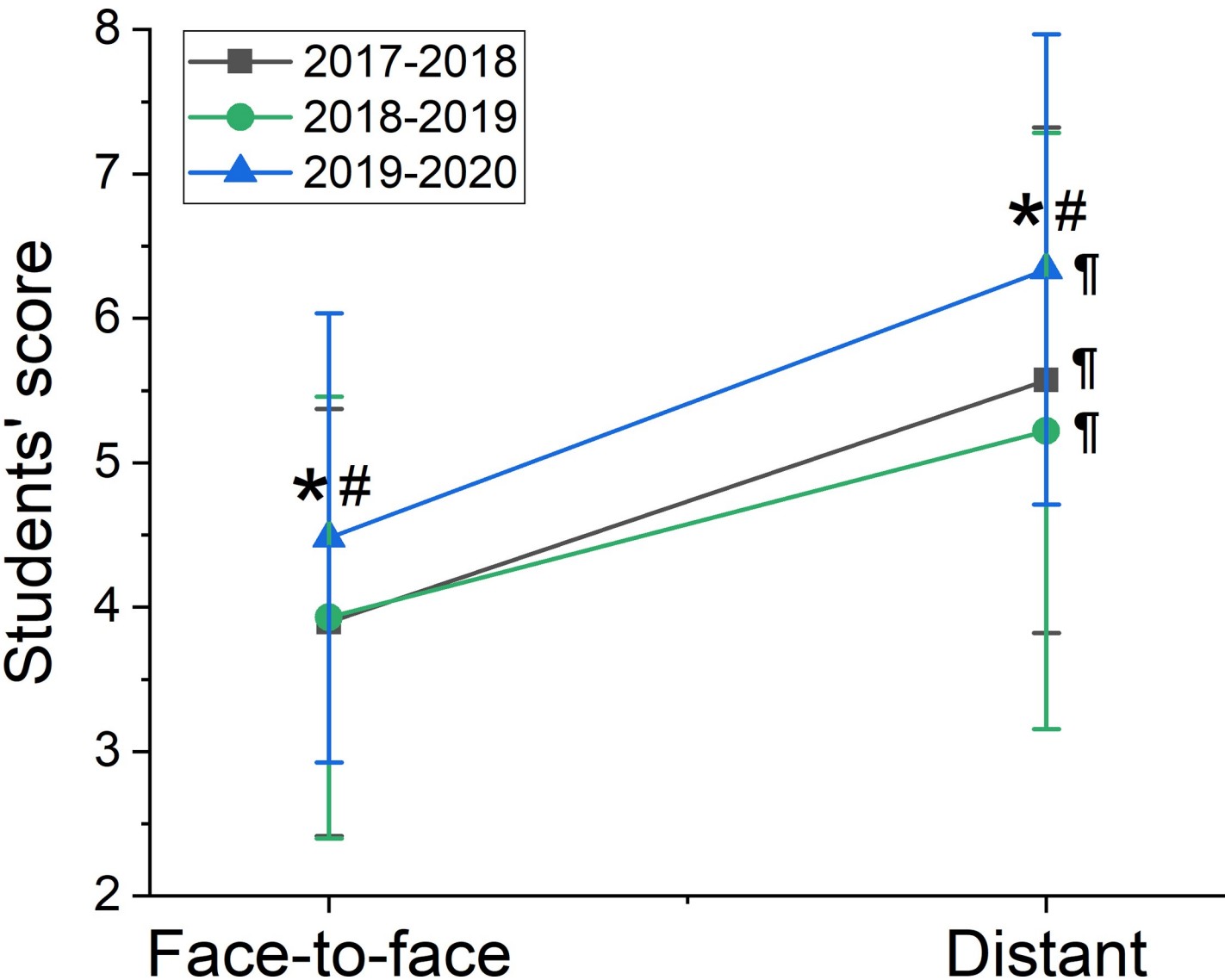

**Fig 7. Students' scores in the face-to-face period (before confinement) and in the distance teaching period (during confinement).** Data is presented for the 3 academic courses (2017/2018 and 2018/2019 being the control group and 2019/2020 the experimental group) as mean±SD. Symbols indicate statistically significant differences (p<0.05) between course 2019/2020 and 2017/2018 (*), 2019/2020 and 2018/2019 (#) or between the period before and after the confinement during the same course (¶).

there were statistically significant differences between the mean of the present course 2019/2020 and the previous 2 years (6.3±1.6 in experimental group, vs. 5.6±1.8 in 2017/2018, p = 0.0168, and vs. 5.2±2.1 in 2018/2019, p = 0.0002), with no differences between 2017/2018 and 2018/2019 (p = 0.4451). The mean scores are significantly different both in the control and in the experimental groups between both periods (before and after confinement). We have also seen an increase in the mean score differences after the confinement.

**3.2.2 Not additional autonomous activities in confinement.** This study has been performed with students from the subject "Metabolism", which is a subject where no additional tasks have been imposed to students because of the COVID-19 confinement. We have analysed students' scores and the proportion of students that passed the activity (score≥5). As it can be observed in Fig 8A, the scores obtained by students in the different activities during the

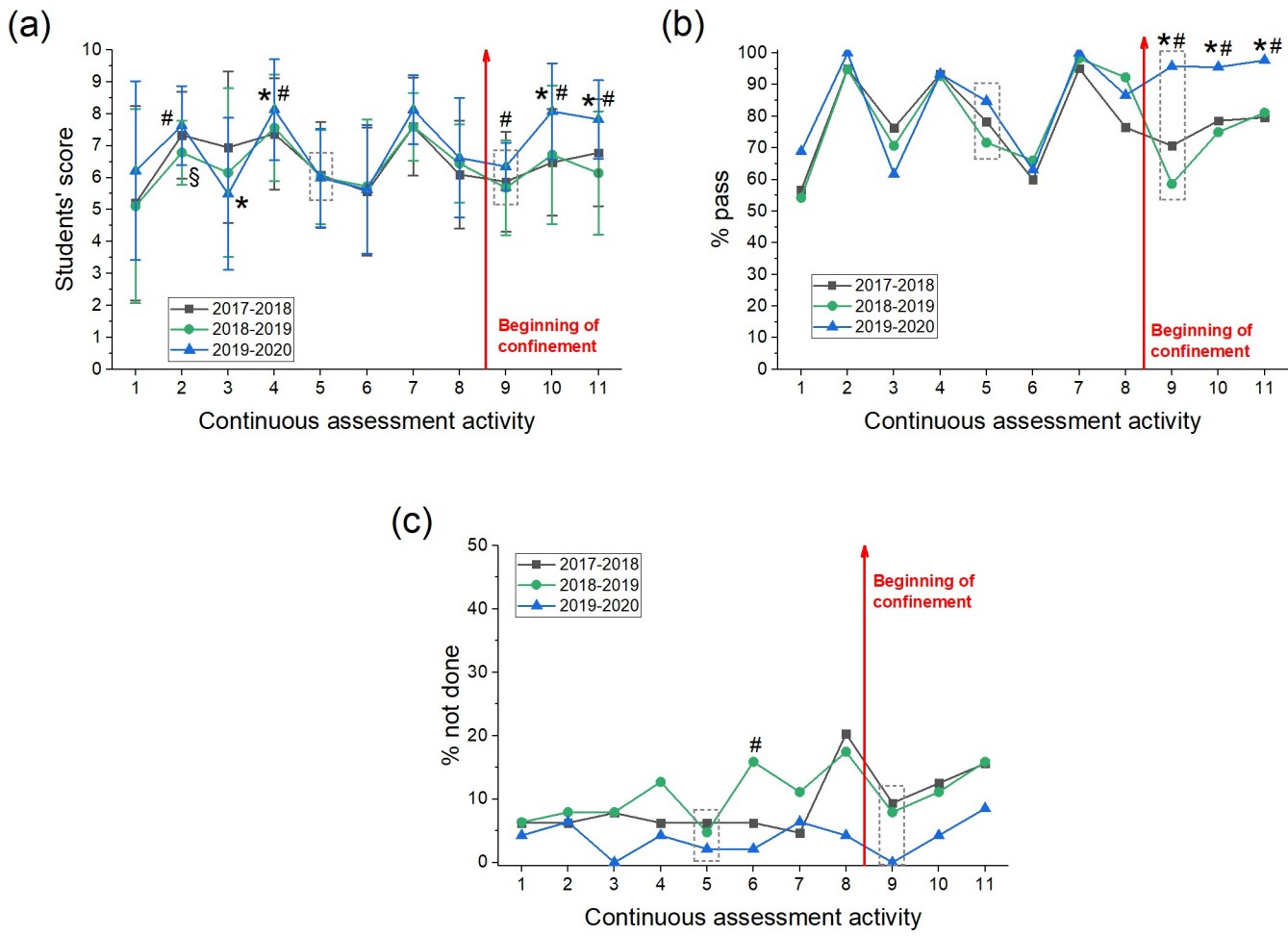

**Fig 8. Results of students enrolled in Metabolism during the last 3 academic years.** (a) The score obtained by students in the different activities of the continuous assessment is represented. (b) Proportion of students that passed the continuous assessment activities (score≥5). (c) Proportion of students that did not carry out the continuous assessment activities. From the 11 activities, only 2 are in-person activities under normal conditions (number 5 and 9, highlighted with a box), the other activities were always on-line. The end of in-person teaching and beginning of confinement is indicated with a red arrow. Data is presented as mean±SD. Symbols indicate statistically significant differences (p<0.05) between courses 2017/2018 and 2018/2019 (§), 2017/2018 and 2019/2020 (*) and 2018/2019 and 2019/2020 (#).

course were consistently similar in both years of the control group, with the only exception in the second activity where there was a slightly decrease in year 2018/2019 (p<0.05). Similarly, the proportion of students with a score≥5 was very similar in the control group (p>0.05) (Fig 8B). Therefore, students' performance in the previous 2 years (control group) was not significantly different, allowing us to compare these results to the results from 2019/2020. Scores obtained by the students in the 2019/2020 academic year before confinement were similar to those obtained by students from the control group, although some differences were found at the beginning of the course (activities number 2, 3 and 4, p<0.05). We think that this could be due to the adaptation of the students to the course and to the new assessment methodology, being afterwards similar again. However, after the end of face-to-face teaching and at the beginning of the confinement, students' scores were significantly higher than in the previous academic years (Fig 8A). The most remarkable differences are evident in activities 10 and 11, that have always been on-line activities. Thus, in activity 10, mean score in the present year (2019/2020) was 8.1±0.2 vs. 6.5±0.2 (p<0.0001) and vs. 6.7±0.3 (p = 0.0005) in 2017/2018 and

2018/2019, respectively. Similarly, in activity 11, the mean score of the experimental group was 7.8±0.2 vs. 6.8±0.2 (p = 0.0014) and vs. 6.1±0.3 (p<0.0001) in 2017–2018 and 2018/2019, respectively. Better performance of students was strongly evident when analysing the proportion of students that passed these 2 activities (Fig 8B). Thus, 95.6% of students passed activity 10 (p<0.0047 vs. 2017/2018 and p = 0.0025 vs. 2018/2019) and 97.7% activity 11 (p<0.0068 vs. 2017/2018 and p = 0.0111 vs. 2018/2019). Moreover, we also found higher scores and an increase in the number of students that passed the activity number 9, which is a test (Fig 8A and 8B). In the present academic course (2019/2020), this test has been performed by the students confined to their homes, in contrast to previous years where the test was performed in the classroom under the supervision of the tutor. In addition, we have analysed the proportion of students that performed the activities in different courses (Fig 8C), finding that this proportion is very similar, with a difference in only one activity.

## 4. Discussion and conclusions

The aim of this study was to identify the effect of COVID-19 confinement on students' performance. Therefore, we conducted an experiment among 450 students from three subjects in different degrees of higher education at Universidad Autónoma de Madrid, Spain. The results of this study answer our 4 research questions:

1.  Is there any effect (positive or negative) of COVID-19 confinement on students' performance?

2.  Is it possible to be sure that COVID-19 confinement is the origin of the different performance (if any)?

3.  What are the reasons of the differences (if any) in students' performance?

4.  What are the expected effects of the differences in students' performance (if any) in the assessment process?

With respect to research question 1, the results analyzed in section 3.2 show that there is a significant positive effect of COVID-19 confinement on students' performance. The results indicate that students obtained better scores in all kinds of tests that were performed after the beginning of confinement. Different sources of error have been removed from our study by including only subjects that fit the following requirements for the last three academic years:

- Same teaching methodology and teachers in all the years.

- Same assessment process in all years, including both distance and on-site activities.

- Tests performed both before and after the COVID-19 confinement in the present academic year.

- Tests that only imply an objective assessment process in order to avoid any possible subjective interpretation by the lecturers.

We have compared results of students from the present 2019/2020 academic year (experimental group) with results of students from the last two academic years 2018/2019 and 2017/2018 (control group). Students from the two previous academic years had similar circumstances from the beginning to the end of their course. When compared to students of the 2019/2020 academic year, we have two periods with different conditions. The first period is taught in the same conditions as in previous years (before confinement). In the second period, those conditions dramatically changed, and all the teaching and learning activities were limited to distance learning. The results of our studies (section 3.2) clearly indicate that there are

significant differences when comparing students' performance in confinement with the performance in previous periods where activities were not limited to distant learning.

At this point, it is clear that the variable that correlates with the change in students' performance is the beginning of the confinement. However, we cannot yet establish if the difference is due to either:

- The new learning methodology, or

- The new assessment process.

For those reasons, we should now answer research question 2 and establish whether the differences found when answering research question 1 are only due to the COVID-19 confinement.

The problem with confinement is that not only the learning, and teaching strategies should be modified, but also the assessment process as it cannot be done face-to-face. There are some concerns related to these new assessment methods such as the opportunity for cheating by the students. This is the reason why we have chosen only subjects that include several tests that have not been modified because of the confinement. The results of our study show that students also achieved significant improvements in their scores even in tests that were performed in the on-line format in previous years. Moreover, this improvement is only significant when comparing data after the confinement (i.e. there are no significant differences in on-line tests that were performed before the confinement). These findings reveal that the new assessment process cannot be the reason for the improvement in students' performance because the learners also achieved better scores when the format of the assessment did not change. For these reasons, we establish that the new learning methodology is the main reason for the change in students' performance during the confinement.

Now, we shall discuss research question 3 (What are the reasons of the differences (if any) in students' performance?). We have proven that something has changed in students' learning methodology. The question is: What is the common element in those methodologies that caused the improvement in the learning process?

We have analyzed data from two different subjects that used two very different learning strategies during the confinement (section 3.2). In one of them (section 3.2.1), additional e-learning tasks were imposed on the students. Theoretical lessons were replaced by written documents. In the other one (section 3.2.2), no additional e-learning tasks were given, but theoretical lessons were replaced by multimedia classes. In both cases, we have found a significant increment in students' performance in the evaluation tests after the confinement. It seems that students' performance is increased independently of the learning strategies followed by teachers. Since we have established that the assessment process cannot be responsible for the differences, and we cannot find any common elements in the learning methodologies, we must conclude on a general change in the autonomous learning process.

We have also analyzed data from the previous years in two subjects that demonstrate that students do not work on a continuous basis (section 3.1). In both subjects, students work hard only in the last days before the final exams. In one subject, we have found that more than 33% of the autonomous work was done in the 5 days immediately preceding the final exam (section 3.1.1). In the other subject, students only used the e-learning material in the last 2 days before the final exam, even when it had been provided three weeks in advance (section 3.1.2). In this study, we have also found that students easily change their learning behavior and study continuously when a reward is offered. This extra motivation dramatically changed their learning strategy and students worked in a much wider time window, increasing their performance. An increase in the students' performance due to adequate time management in the learning process is well-established in the literature [12, 13].

In the present COVID-19 confinement, students can find by themselves many different motivations (rewards) to work on a continuous basis. First, the confinement is a new scenario that has never been faced by the students. For this reason, students do not have any previous experience to use as a reference in their learning process. Without any previous reference, students need to be confident that they are following the course correctly and therefore, work continuously in order not to miss any important content. Another interpretation is that they are afraid of missing the academic year because of the COVID-19 confinement and they work harder to overcome any difficulty. Finally, students might be motivated by their intrinsic responsibility in a very confused situation and work hard to contribute as much as they can to solve the problems that higher education is facing. Most probably, different students will find different motivations in this new scenario (probably a combination of many). We conclude that there is a real and measurable improvement in the students' learning performance that we believe can guarantee good progress this academic year despite the COVID-19 confinement.

Answering question 4, we have demonstrated that students get better grades in activities that did not change their format after the COVID-19 confinement. Moreover, we have demonstrated that there is an improvement in their learning performance. In conclusion, higher scores are expected due to the COVID-19 confinement that can be directly related to a real improvement in students' learning.

## Supporting information

**S1 Data.**
(XLSX)

## Acknowledgments

The authors want to thank Dr. Gema Perez-Chacón for her valuable comments on statistical analysis.

## Author Contributions

**Conceptualization:** Santi Fort.

**Formal analysis:** T. Gonzalez, K. P. Hincz, M. Comas-Lopez, Laia Subirats, G. M. Sacha.

**Investigation:** T. Gonzalez, M. A. de la Rubia, M. Comas-Lopez, Laia Subirats, G. M. Sacha.

**Methodology:** T. Gonzalez, Laia Subirats.

**Project administration:** Santi Fort, G. M. Sacha.

**Resources:** M. A. de la Rubia.

**Software:** K. P. Hincz.

**Supervision:** M. A. de la Rubia, Santi Fort, G. M. Sacha.

**Validation:** Laia Subirats.

**Writing – original draft:** T. Gonzalez, G. M. Sacha.

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
