## [Decision Letter · Decision Letter 0]

16 Jun 2020

PONE-D-20-11966

Influence of COVID-19 confinement in students’ performance in higher education

PLOS ONE

Dear Dr. Gomez-Monivas,

Thank you for submitting your manuscript to PLOS ONE. After careful consideration, we feel that it has merit but does not fully meet PLOS ONE’s publication criteria as it currently stands. Therefore, we invite you to submit a revised version of the manuscript that addresses the points raised during the review process.

We look forward to receiving your revised manuscript.

Kind regards,

Haoran Xie

Academic Editor

PLOS ONE

Journal Requirements:

Reviewers' comments:

Reviewer's Responses to Questions

**Comments to the Author**

1. Is the manuscript technically sound, and do the data support the conclusions?

Reviewer #1: Yes

Reviewer #2: Yes

2. Has the statistical analysis been performed appropriately and rigorously? 

Reviewer #1: Yes

Reviewer #2: Yes

3. Have the authors made all data underlying the findings in their manuscript fully available?

Reviewer #1: No

Reviewer #2: Yes

4. Is the manuscript presented in an intelligible fashion and written in standard English?

Reviewer #1: No

Reviewer #2: Yes

5. Review Comments to the Author

Reviewer #1: The organization of the resutls section and the addressed questions (1-4) is a bit unclear please reorganize this so it becomes clear what analysis has been used to address what question.

Abstract:

This study explores the effects of COVID-19 confinement in the students’ performance in higher education. This sentence should be rephrased more clearly without genitive form.

The language of the whole paper should carefully revised. At the moment the paper is hard to understand.

In this subject we have analyzed students’ performance expressed... I needed to guess that the name of the subject is 'applied computing'. This needs to be expressed more clearly, also in the section headings: For instance:

2.3.1 Subject: Applied computing

Question 2: The results of our studies clearly indicate that there are significant differences in students’

519 performance after the confinement that cannot be found before in the same year or when comparing to the previous

520 academic years. It is unclear how this has been established statistically. Please clarify.

Where is the Wilcox test for Fig 7 comparing the academic years?

I missed how the following has been studied: How do you exclude the possibility that the lecturers just gave better scores? Given the fact that the university would not like to see worse grade of the students because this could reflect negatively on the institution and their management this is a valid question. Please address it.

Reviewer #2: Aims of the study were clearly defined and followed, data were appropriately collected and analysed, results were adequately discussed, and conclusions were well supported. I propose the following minor revisions.

1.The first part of the “introduction” focuses on the assessment, which is not appropriate given the main purpose of the study is to investigate the influence of COVID-19 confinement. I would recommend to start directly with “1.1 background” with a brief introduction of the e-learning in the situation of COVID-19.

2.It would be much clearer if the section “participant” is placed before the section of “statistical analysis”.

3.The results presented in figures 4-6 are not reader-friendly.

6. PLOS authors have the option to publish the peer review history of their article (what does this mean?). If published, this will include your full peer review and any attached files.

Reviewer #1: No

Reviewer #2: No

---

## [Author Response · Author response to Decision Letter 0]

25 Jul 2020

Reviewer 1:

The organization of the results section and the addressed questions (1-4) is a bit unclear please reorganize this so it becomes clear what analysis has been used to address what question.

We have included explicit references to the analysis that has been used to answer each research question. We have also clarified the point of the discussion that jumps from answering research question 1 to research question 2, which was the most confusing part of the discussion.

Abstract:

This study explores the effects of COVID-19 confinement in the students’ performance in higher education. This sentence should be rephrased more clearly without genitive form.

That sentence has been rephrased as reviewer suggested.

The language of the whole paper should carefully revised. At the moment the paper is hard to understand.

We have carefully revised the language in the whole paper.

In this subject we have analyzed students’ performance expressed... I needed to guess that the name of the subject is 'applied computing'. This needs to be expressed more clearly, also in the section headings: For instance:

2.3.1 Subject: Applied computing

We have changed the order of the information in section 2. Now, the description of the subjects is placed before the statistical analysis. In the subsection participants, we have also included a sentence that explicitly indicates the name of the subjects, that are described right after that sentence.

Question 2: The results of our studies clearly indicate that there are significant differences in students’

519 performance after the confinement that cannot be found before in the same year or when comparing to the previous

520 academic years. It is unclear how this has been established statistically. Please clarify.

The reviewer is right when asking about clarification in this sentence because the sentence was not correctly written. In the present version, it reads:

“The results of our studies (section 3.2) clearly indicate that there are significant differences when comparing students’ performance in confinement with the performance in previous periods where activities were not limited to distant learning.”

This is one of the main results of the article and it has been established statistically in section 3.2

Where is the Wilcox test for Fig 7 comparing the academic years?

Data from this subject imply results from an adaptive test. This imply that there are some points in the test where students get stacked and some scores are repeated more often than others. For this reason, data did not pass the normality test, as we explain in section 2.4.1. We have used non-parametric statistical tests such as Kruskal-Wallis and Mann-Whitney post hoc test, as explained in the same section. Results including the statistical analysis are included in the third paragraph of section 3.2.1. (starting in line 452).

We have extended the explanation in section 2.4.1. 

I missed how the following has been studied: How do you exclude the possibility that the lecturers just gave better scores? Given the fact that the university would not like to see worse grade of the students because this could reflect negatively on the institution and their management this is a valid question. Please address it.

Thank you for asking that important question, that must be clarified in the article indeed. The subjects involved in this article mainly include tests that are objective, such as multiple answer tests or open questions with a very specific and quantitative answer that cannot be altered by the teachers. For that reason, we are sure that it is impossible for lecturers to give better scores.

We have included a point that indicates this fact in the discussion and conclusions section.

Reviewer 2:

Aims of the study were clearly defined and followed, data were appropriately collected and analysed, results were adequately discussed, and conclusions were well supported. I propose the following minor revisions.

1.The first part of the “introduction” focuses on the assessment, which is not appropriate given the main purpose of the study is to investigate the influence of COVID-19 confinement. I would recommend to start directly with “1.1 background” with a brief introduction of the e-learning in the situation of COVID-19.

The first part of the introduction, (section 1) talks in general terms about the problem studied in the article. Then, in section 1.1 we give more details about several topics such as w-learning, Self-regulated learning and adaptive tests. For that reason, we do not think it is adequate to completely remove section 1 and start directly in section 1.1. however, reviewer comments are right in the sense that, maybe the part that talks about assessment is too long and an introduction about e-learning in confinement is required.

For those reasons, we have reduced the assessment discussion in section 1.1 and a brief introduction of the e-learning in the situation of COVID-19 has been added in section 1.1 Background.

2.It would be much clearer if the section “participant” is placed before the section of “statistical analysis”.

We have modified the order of these subsections as reviewer suggested.

3.The results presented in figures 4-6 are not reader-friendly.

Figures have been modified to increase readability by increasing the size of the smaller letters. In figure 6, we have changed even the structure of the figure to make it more reader-friendly.

Additional changes

1. Changed the word “significative” for “significant” in 5 places

2. Notation changes in the section “2.1.1 CAT theoretical model” (some changes of psi by fi)

3. The affiliation Universitat Oberta de Catalunya has been added to the author Laia Subirats

4. Affiliations have been renumbered by order of appearance

---

## [Decision Letter · Decision Letter 1]

8 Sep 2020

Influence of COVID-19 confinement on students’ performance in higher education

PONE-D-20-11966R1

Dear Dr. Gomez-Monivas,

We’re pleased to inform you that your manuscript has been judged scientifically suitable for publication and will be formally accepted for publication once it meets all outstanding technical requirements.

Kind regards,

Haoran Xie

Academic Editor

PLOS ONE

Additional Editor Comments (optional):

Reviewers' comments:

Reviewer's Responses to Questions

**Comments to the Author**

1. If the authors have adequately addressed your comments raised in a previous round of review and you feel that this manuscript is now acceptable for publication, you may indicate that here to bypass the “Comments to the Author” section, enter your conflict of interest statement in the “Confidential to Editor” section, and submit your "Accept" recommendation.

Reviewer #1: All comments have been addressed

Reviewer #2: All comments have been addressed

2. Is the manuscript technically sound, and do the data support the conclusions?

Reviewer #1: Yes

Reviewer #2: Yes

3. Has the statistical analysis been performed appropriately and rigorously? 

Reviewer #1: Yes

Reviewer #2: Yes

4. Have the authors made all data underlying the findings in their manuscript fully available?

Reviewer #1: No

Reviewer #2: Yes

5. Is the manuscript presented in an intelligible fashion and written in standard English?

Reviewer #1: Yes

Reviewer #2: Yes

6. Review Comments to the Author

Reviewer #1: All comments have been addressed.

Please use the space provided to explain your answers to the questions above. You may also include additional comments for the author, including concerns about dual publication, research ethics, or publication ethics. (Please upload your review as an attachment if it exceeds 20,000 characters) (Limit 100 to 20000 Characters)

Reviewer #2: My comments have been considered and addressed. I have no more objection on the acceptance of the manuscript.

7. PLOS authors have the option to publish the peer review history of their article (what does this mean?). If published, this will include your full peer review and any attached files.

Reviewer #1: No

Reviewer #2: No

---

## [Editor Report · Acceptance letter]

29 Sep 2020

PONE-D-20-11966R1 

Influence of COVID-19 confinement on students’ performance in higher education 

Dear Dr. Sacha:

I'm pleased to inform you that your manuscript has been deemed suitable for publication in PLOS ONE. Congratulations! Your manuscript is now with our production department. 

Kind regards, 

on behalf of

Professor Haoran Xie 

Academic Editor

PLOS ONE